# Improving the Accuracy of Coal-Rock Dynamic Hazard Anomaly Detection Using a Dynamic Threshold and a Depth Auto-Coding Gaussian Hybrid Model

Yulei Kong and Zhengshan Luo *

School of Management, Xi'an University of Architecture and Technology, Xi'an 710055, China
* Correspondence: luozhengshan@163.com

**Abstract:** A coal-rock dynamic disaster is a rapid instability and failure process with dynamic effects and huge disastrous consequences that occurs in coal-rock mass under mining disturbance. Disasters lead to catastrophic consequences, such as mine collapse, equipment damage, and casualties. Early detection can prevent the occurrence of disasters. However, due to the low accuracy of anomaly detection, disasters still occur frequently. In order to improve the accuracy of anomaly detection for coal-rock dynamic disasters, this paper proposes an anomaly detection method based on a dynamic threshold and a deep self-encoded Gaussian mixture model. First, pre-mining data were used as the initial threshold, and the subsequent continuously arriving flow data were used to dynamically update the threshold to solve the impact of artificially setting the threshold on the detection accuracy. Second, feature dimensionality reduction and reorganization of the data were carried out, and low-dimensional feature representation and feature reconstruction error modeling were used to solve the difficulty of extracting features from high-dimensional data in real time. Finally, through the end-to-end optimization calculation of the energy probability distribution between different categories for anomaly detection, the problem that key abnormal information may be lost due to dimensionality reduction was solved and accurate detection of monitoring data was realized. Experimental results showed that this method has better performance than other methods.

**Keywords:** coal-rock dynamic hazard; depth self-coding Gaussian mixture model; dynamic threshold; anomaly detection algorithm

## 1. Introduction

As the basic energy source of the national economy, coal has made great contributions to the economic development of all countries in the world in the recent 100 years. With the coming of the carbon emission reduction era, the use of coal will gradually reduce, but until 2050, coal is still the most economic main energy source. As shallow coal is close to depletion after long-term mining, deep mining has become the norm, and with the increasing depth and intensity of coal mining, coal-rock power disasters are increasing, which creates a substantial threat to the safety of enterprise property and personnel life.

Many coal mines adopt different mining techniques according to the different forms of coal seams. Take upward mining as an example; upward mining is generally affected by multiple factors, including but not limited to the complex integrity of the interlayer rock strata affected by the disturbance of the mining of the lower coal seam, the accumulation of gas and harmful gases in the underlying goaf, and the occurrence of abnormal mine pressure in the upward mining face. Coal and rock dynamic disasters occur due to downward mining. Therefore, early and timely detection of abnormal changes in the coal and rock mass can prevent the occurrence of disasters. Thus, improving the accuracy of coal-rock dynamic disaster abnormality detection has become a key research direction and a hot spot for the majority of scholars. Yuan proposed the theory of intelligent discrimination of coal-rock dynamic disaster multiparameter precursor information and an early warning model

and developed a precursor information acquisition sensing equipment and supporting technology with fault self-diagnosis and high sensitivity [1]. Wang made a study of the recent years of disaster monitoring research. The current status of coal-rock power disaster monitoring and early warning research was summarized, and the commonly used methods are mainly conventional static prediction, index prediction, geophysical monitoring, and mathematical model prediction [2]. Li proposed a rock failure early warning method by introducing the Hurst exponent into the geotechnical field to reflect the long memory and fractal structure of the time series; the improved R/S method proposed overlapping subsequences, so the calculation of the Hurst index is better. At the same time, using the Hurst index supplemented by acoustic emission/microseismic activity monitoring can predict early warning points and improve disaster prevention [3].

Pang summarized the application of deep learning in anomaly detection and divided deep anomaly detection methods into three conceptual paradigms from a modeling perspective: deep feature extraction, mean feature learning, and end-to-end anomaly scoring [4]. Bulusu provided a comprehensive comparison of the relative advantages and disadvantages of supervised, semi-supervised, and unsupervised and some other anomaly detection methods, arguing that the unsupervised technique, due to its lack of dependence on labeled data, is more suitable for anomaly detection applications [5]. Hojjati summarized the self-supervised anomaly detection methods [6] and tested them against the widely used deep shallow and generative models. Self-supervised anomaly detection algorithms (SSL) significantly outperformed other algorithms, and this advantage makes self-supervised algorithms a key branch of anomaly detection. Zhao developed a reliability-based design optimization (RBDO) method to determine engineering design parameters by combining the method of moments and high-dimensional model representation (HDMR) and to realize the complex, high-dimensional, and nonlinear response of engineering structures for the surrounding. It is a simple and feasible method to consider uncertainties in rock engineering design, stability analysis, and production [7].

Although research on anomaly detection has touched upon various perspectives, due to the complexity of the geological environment and the high level of difficulty of data processing, this worldwide problem remains unsolved so far. One of the issues of anomaly detection is that it is difficult to distinguish the boundary between normal and abnormal states; the second difficulty is that the threshold value for disaster occurrence varies across mining environments, and the traditional detection method uses historical data as the threshold value, which often results in disasters not occurring beyond the threshold value but occurring within the threshold value. The third difficulty is that the monitoring data arrive in a stream, which can only be read once and require real-time extraction of abnormal features, otherwise the arrival of subsequent data causes the previously processed data to lose their value due to obsolescence.

To address the aforementioned problems, this paper analyzes the anomaly detection processes one by one and finds that the threshold setting method and the anomaly information mining method are the key factors affecting detection accuracy; accordingly, a new anomaly detection algorithm DT-DAGMM is proposed. The DT-DAGMM uses dynamic threshold and deep autoencoding Gaussian mixture model techniques, which not only can better handle high-dimensional data but also can solve the uncertainty of manually set thresholds. It has excellent ability to continuously detect input stream data in real time in the unsupervised state. Specifically, the main contributions are the following three points:

1. The proposed dynamic threshold setting method solves the problem of manual determination of thresholds, affecting the accuracy of anomaly detection.
2. The application of a dynamic threshold fused with the depth self-coding Gaussian mixture model improves the data processing speed and detection accuracy.
3. The coal-rock dynamic disaster monitoring and early warning system is improved.

## 2. Related Work

Over the past decades, anomaly detection techniques have evolved and many research results have been achieved. Among them, unsupervised anomaly detection methods based on deep learning are prominent in many fields, which provide many easy-to-use network structure frameworks that can handle any data type, learn complex structural relationships from different types of data, and improve data usage and detection performance through end-to-end optimization. The four main types of unsupervised deep anomaly detection commonly used today are feature reconstruction, clustering, density-based methods, and single classification methods.

The methods based on feature reconstruction mainly include principal component analysis (PCA) [8], robust principal component analysis (ROBPCA) [9], structured sparse learning [10], and a Deep Auto-encoder [11]. Among them, principal component analysis transforms component-related original variables into component-unrelated new composite variables with the help of orthogonal transformation, revealing as much as possible the original inter-data relationships. Liu et al. tried to combine PCA and the DAGMM and proposed the PCA–DAGMM unsupervised anomaly data detection method [12]. However, principal component analysis is sensitive to outliers and noisy data, so robust PCA was introduced to solve the problem of anomalous noise points. Structured sparse learning, in contrast, modifies the penalty term based on the standard sparse algorithm to force the features to be arranged according to rules, while a deep autoencoder uses the input data themselves as supervision to guide the neural network to learn the mapping relationship to obtain the reconstructed output, and the original time series is considered anomalous when the difference between the reconstructed output and the original input exceeds a specified threshold. Zou et al. used a hybrid autoencoder to replace the single deep autoencoder to generate tandem low-dimensional representations to improve the accuracy of anomaly detection for high-dimensional data [13].

The principle of the density-based anomaly detection method is to partition the data space, and the class cluster density of the region where the anomalous sample points are located is lower than the class cluster density where the normal sample points are located. Li et al. proposed a density-based anomaly data detection algorithm [14], which introduces a sliding time window to prune and filter the data using a grid and then uses outlier factors to make a final judgment on the remaining data, which can improve detection accuracy and data efficiency.

Clustered anomaly detection methods use neural networks to encode the input data into different clusters, and the anomalous data do not belong to any of the clusters. DB-SCAN divides high-density regions into clusters and finds clusters of arbitrary shapes in the noise space. DAE-DBC uses an autoencoder for dimensionality reduction and then identifies outliers by clustering. The cluster function adds L2 normalization as well as k-means to the autoencoder. The gmm function fits arbitrarily shaped data distributions by finding mixed representations of multidimensional Gaussian model probability distributions. The clustering-based approach reduces the dimensionality before clustering, which tends to lose key information in the process of dimensionality reduction. Wang et al. introduced an unsupervised deep clustering framework to better model the representation distribution by adjusting the Gaussian components and improve the intra-cluster compactness and inter-cluster separability through training [15]. Based on the non-dominated sorting genetic algorithm, the multi-objective optimization of the sound transmission of a multilayer composite cylindrical shell lined with a porous core was measured to obtain the internal conditions of the material by measuring the changes generated by the lined porous material receiving plane acoustic waves [16]. Talebitooti et al. also studied the effect of porous material properties on the acoustic transmission of the sandwich aerospace composite hyperbolic shell diffusion field [17].

The goal of the single classification algorithm [18] is to determine whether a query object belongs to the class observed during training. After years of evolution, recent trends in single classification methods have focused on the development of deep-learning-based

approaches. The Old Gold Net (OGN) algorithm [19] developed by Zaheer et al. consists of a generator network and a discriminator, using the generator to generate two images to simulate normal and abnormal inputs and the discriminator to distinguish abnormal from normal. The progressive knowledge refinement method [20] proposed by Zhang et al. trains two networks on a given class of training data, initialized to the same architecture and then trained using reconstruction loss.

Deep learning anomaly detection methods are also well applied in processing sensor data. A comprehensive analysis of the applications by Mohammadi et al. [21] concluded that the main methods applicable to IoT sensing systems are the deep autoencoder (DAE) [22], the deep belief network (DBN) [23], and the long short-term memory (LSTM) network [24]. Among them, the DAE is a data compression algorithm that improves the efficiency of feature extraction by compressing and reducing the dimensionality of unlabeled datasets. The disadvantage is the long training time layer by layer. The component of the DBN is the restricted Boltzmann machine (RBM), which consists of explicit and hidden layer neurons; the explicit layer is used to receive input, and the hidden layer is used to extract features. The layer-by-layer training method is used to solve the optimization problem of deep neural networks, which gives better initial weights to the whole network using layer-by-layer training, but there is still the problem of a long training time layer by layer. The LSTM network adopts the gate structure control mode of a forgetting gate, an input gate, and an output gate to process sequence data and judge whether the information is useful or not, with the drawback that the model structure is complex and has the disadvantages of parallel processing and time-consuming computation.

For different application areas, different detection methods need to be used and the choice of models needs to be determined using the nature of the input data. The data collected by sensors for monitoring coal-rock dynamic hazards are time-series stream data and have different characteristics than general data: A wide variety of data collection devices are deployed in various regions and continuously generate a large number of data streams; various sensing devices collect different information that leads to the complexity of heterogeneous data from multiple sources; and the data collected by sensing devices are spatiotemporally correlated, so each data item has a spatial location and a time stamp. Traditional detection techniques are not capable of processing data with temporal stream characteristics, so methods with the ability to process high-speed data streams in real time are needed.

Summarizing the advantages and disadvantages of the aforementioned types of algorithms, all of them suffer from the shortcomings of manual setting of thresholds and a complex and time-consuming training process, which cannot meet the real-time requirements of coal-rock dynamic hazard anomaly detection. Inspired by these research results, we propose a dynamic threshold deep autoencoding Gaussian mixture model (DT-DAGMM), which is different from the existing models: the model uses adaptive dynamic thresholding instead of manual setting to improve the accuracy and detection speed of anomaly detection and uses an evaluation network to train the reconstruction error and feature extraction together to reduce the loss of information from dimensionality reduction and make the model more effective.

## 3. DT-DAGMM Anomaly Detection Method

The DT-DAGMM anomaly detection method proposed in this paper is implemented through models and algorithms, and the design goal is to be able to detect anomalous dynamics in real time and accurately, so the design of the models and algorithms needs to meet the following requirements:

1. Adaptive dynamic thresholds are set using streaming data processing methods.
2. The entire data must be read in one stream.
3. The current $tx$ must be identified as normal or abnormal before receiving subsequent $tx + 1$ data.
4. There is no need to use labeled data and manual parameter adjustment.



The DT-DAGMM constructed according to the research objectives and requirements of this paper comprehensively represents the process and functions of each link of anomaly detection.

### 3.1. DT-DAGMM

The objective of the construction of the DT-DAGMM is to enable the streaming processing of sensing monitoring data and the real-time detection of disaster anomalies. As shown in Figure 1, the DT-DAGMM consists of three modules: dynamic threshold determination, data compression, and depth anomaly detection. It works as follows: (1) The dynamic threshold module uses the stream data collected in the unexploited state as the initial threshold and dynamically updates the threshold using the stream data that arrive later to eliminate the dependence on subjective determination of the threshold; (2) the data compression module performs downscaling and feature reconstruction of the input data through a depth autoencoder to discover more representative features of the disaster; and (3) the depth anomaly detection module combines the low-dimensional feature representation and reconstruction error input from the data compression module for modeling and performs dynamic detection of coal-rock dynamic disaster anomalies by calculating the energy probability distribution among different categories within a Gaussian mixture framework.

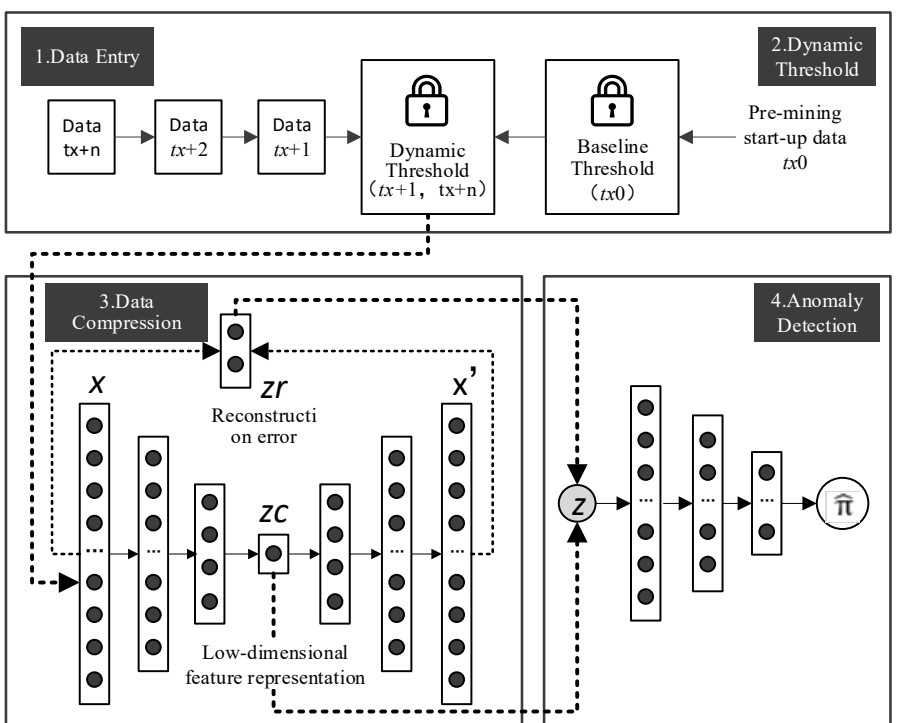

**Figure 1.** DT-DAGMM coal-rock dynamic disaster real-time anomaly detection method.

The DT-DAGMM has three advantages: The first is that it does not require manual setting of thresholds and can dynamically update the threshold adaptively, which reduces subjectivity and uncertainty and makes the anomaly detection results more accurate. The second is that it can process high-dimensional data in real time, and finally, it can adapt to different data types and scenarios through different feature representations and hybrid model parameters, which has good generality and adaptability.

### 3.2. Dynamic Threshold Determination

The dynamic threshold is a threshold algorithm that can be dynamically updated and adaptively adjusted. The threshold value, as the demarcation standard between normal

and abnormal, is directly related to the accuracy of abnormality detection. A high threshold value will make normal and abnormal indistinguishable, and a low threshold value will cause a false alarm. It is a common practice to use manually determined thresholds for coal-rock dynamic hazard anomaly detection. This method ignores the tectonic variability of different geological environments, and the set thresholds are often too high or too low, which directly affects the accuracy of anomaly detection. An ideal real-time anomaly detection algorithm should be able to determine whether new arrivals are anomalous, without relying on pre-set thresholds. The basic principle of dynamic threshold design is that $tx$ data must be identified as normal or abnormal before receiving subsequent $tx$ +1 data. In this paper, the initial data before mining are used as the threshold benchmark and the threshold is dynamically updated using subsequent successive arrivals of stream data, which not only improves the accuracy of the threshold but also adapts to different monitoring environments. The dynamic threshold is determined using the following equation:

$$\theta = tx_0, \ \{(TX = tx_0, tx_1, \cdots, tx_n), (X = 0, 1)\}, \tag{1}$$

where $\theta$ is the threshold value, based on the intermediate data $tx_0$ in the zero-start state with the time index loaded; $TX$ is the streamed data over time; $X$ is the sum of data, consisting of 0 and 1; $x_0$ represents normal data; and $x_1$ represents abnormal data.

### 3.3. Feature Reduction and Restructuring

The purpose of feature downscaling and reorganization is to reduce the amount of data computation and extract more typical and representative features.

Two types of anomalies generally occur in monitoring data: One is anomalies not related to catastrophes, such as insufficient battery capacity and equipment failure. The other type is disaster anomalies. The data from a single sensor cannot truly reflect which kind of abnormality the monitoring object belongs to, and only through comprehensive continuity analysis of multi-sensor data can an accurate judgment be made. At present, the three-step method of data acquisition, data pre-processing, and data classification is commonly used for feature extraction; the three-step method of statistical, format, and topological features is also used for feature extraction. All these methods have many pain points: One is that it is difficult to distinguish two kinds of anomalies, and the other is that the processing is tedious and far beyond the time limit requirement of monitoring data processing.

A deep autoencoder can automatically learn features from unlabeled data and give better feature descriptions than the original data and is therefore applied in this paper for feature dimensionality reduction and reorganization. The architecture of a DAE consists of two core components: an encoder and a decoder. The encoder encodes the high-dimensional input into a low-dimensional hidden variable that allows the neural network to learn the most typical features, and the decoder reduces the hidden variable in the hidden layer to dimensionality at the time of input, i.e., input $\approx$ output. The feature downscaling restructuring consists of two elements: first, the recoding and nonlinear downscaling of high-dimensional data using the DAE, and second, the reconstruction of feature elements to obtain a more typical description and interpretation of the monitoring target. The specific algorithm is as follows: The input high-dimensional data $x$ are compressed and downscaled by the encoder several times to obtain a more comprehensive and representative low-dimensional feature set $z_c$, $z_c$ is reconstructed by the decoder to obtain the reconstructed feature $x'$, the reconstructed error $z_r$ is formed between $x$ and $x'$, and the combination of the low-dimensional representation $z_c$ and the reconstructed error $z_r$ constitutes $z$, which provides low-dimensional feature representation and reconstructed error information for subsequent modeling and detection. The feature dimensionality reduction and reconfiguration equations are shown below:

$$z_c = h(x; \theta_e), \ x' = g(z_c; \theta_d), \tag{2}$$

$$z_r = f(x, x'),\tag{3}$$

$$\mathbf{z} = [z_c, z_r],\tag{4}$$

where $z_c$ is the low-dimensional representation, $z_r$ is the reconstruction error, $\theta_e$ and $\theta_d$ are parameters, $x'$ is the reconstruction of $x$, $h(\cdot)$ is the encoding function, $g(\cdot)$ is the decoding function, $f(\cdot)$ is the function that calculates the reconstruction error characteristics, and $z$ is the combination of the low-dimensional representation $z_r$ and the reconstruction error $z_r$.

### 3.4. Deep Anomaly Detection

Since the key anomalous information may be lost during downscaled feature extraction, further low-dimensional spatial density estimation of the feature combination in the Gaussian mixture model is required. As shown in Figure 1, the input z of the estimation network comes from the low-dimensional representation synthesized by the depth self-encoder $z_c$ and $z_r$. The number of classes is assumed to be k in the Gaussian mixture model, and the output p is obtained by the classifier as the probability that the samples belong to k classes, and then the probability is used to estimate the GMM mean, variance, and covariance matrix parameters to obtain the sample energy in the DAGMM objective function. The samples are judged to be anomalous according to the energy probability values. Given the low-dimensional representation z and the integer K as the number of mixed components, the affiliation prediction formula is as follows:

$$P = MLN(Z; \theta_m), \hat{\gamma} = \text{softmax}(P),\tag{5}$$

where P is the output of the multilayer network parameterized by $\theta_m$. $\hat{\gamma}$ is the k-dimensional vector used for the soft mixture component affiliation prediction. Given the sample and its affiliation prediction, $\forall \le k \le K$, we further estimate the parameters in the GMM as follows:

$$\widehat{\phi_k} = \sum_{i=1}^{N} \frac{\hat{\gamma}_{ik}}{N}, \ \hat{\mu}_k \frac{\widehat{\sum}_{i=1}^{N} \hat{\gamma}_{ik} z_i}{\sum_{i=1}^{N} \hat{\gamma}_{ik}}, \ \widehat{\sum}_k = \frac{\sum_{i=1}^{N} \hat{\gamma}_{ik}(z_i - \hat{\mu}_k)(z_i - \hat{\mu}_k)^T}{\sum_{i=1}^{N} \hat{\gamma}_{ik}},\tag{6}$$

where $\hat{\gamma}$ is the predicted affiliation of the low-dimensional representation $z_i$, $\widehat{\varnothing}_k$ is the mixture probability of the $k$-component, $\hat{\mu}_k$ is the mean, and $\widehat{\sum}_k$ is the covariance. The estimated parameters are used to further infer the energy function:

$$E(z) = -\log\left(\sum_{k=1}^{K} \hat{\phi}_k \frac{\left(-\frac{1}{2}(z - \hat{\mu}_k)T\widehat{\sum_K^{-1}}(Z - \hat{\mu}_K)\right)}{\sqrt{\left|2\pi\widehat{\sum_K}\right|}}\right),\tag{7}$$

where $|\cdot|$ denotes the determinant of the matrix.

Compared with the commonly used coal-rock dynamic hazard anomaly detection algorithm, this method can better handle high-dimensional data and solve the uncertainty of manually set thresholds.

## 4. Experiment and Analysis

### 4.1. Experimental Setup

The experiments were based on Windows 10 OS, the Intel(R) Core(TM) i7-10700F CPU @ 2.90 GHz processor, 64.0 GB RAM, and the TensorFlow deep learning framework [25].

1.  Dataset: The UCI machine learning library KDDCUP [26] and the KDDCUP-Rev dataset were used.
2.  Evaluation metrics: The average precision, recall, and F1-measure were used as evaluation metrics.

3. Baseline comparison models: The OC-SVM [27] is a commonly used classical anomaly detection method using the radial basis function (RBF) kernel technique. DSEBM-e [28] is a state-of-the-art unsupervised anomaly detection deep learning method based on the DSEBM (deep structural energy model); DSEBM-r is the same as the DSEBM-e core technology, but DSEBM-r uses the reconstruction error as the criterion for anomaly detection. The deep clustering network (DCN) [29] is a state-of-the-art clustering algorithm that regulates the self-encoder performance using k-means and uses the distance between the instance and the cluster center as the criterion for anomaly detection; the more the distance is from the center, the more likely the instance is anomalous.

4. Parameter setting: See Table 1 for details.

**Table 1.** Parameter settings for the experimental dataset.

| Dataset | Sample Size | Data Dimension | $\lambda_1$ | $\lambda_2$ | Anomaly Ratio |
|---|---|---|---|---|---|
| KDDCUP | 494.021 | 120 | 0.1 | 0.005 | 0.2 |
| KDDCUP-R$_{ev}$ | 121.597 | 120 | 0.1 | 0.005 | 0.2 |

The KDDCUP compression network provides 1 low-dimensional and 2 reconstruction error inputs for the estimation network, using FC(120, 60, tanh)-FC(60, 30, tanh)-FC(30, 10, tanh)-FC(10, 1, none)-FC(1, 10, tanh)-FC(10, 30, + tanh)-FC(30, 60, tanh)-FC(60, 120, none) runs; the estimation network provides 1 GMM containing 4 mixed components, using FC(3, 10, tanh)-Drop(0.5)-FC(10, 4, softmax) runs.

The KDDCUP-Rev compression network also provides 1 low-dimensional and 2 reconstruction error inputs, using FC(120, 60, tanh)-FC(60, 30, tanh)-FC(30, 10, tanh)-FC(10, 1, none)-FC(1, 10, tanh)-FC(10, 30, tanh)-FC(30, 60, tanh)-FC(60, 120, none) runs, and the estimated network provides GMMs containing 2 mixed components, using FC(3, 10, tanh)-Drop(0.5)-FC(10, 2, softmax) runs.

Here, FC(a, b, f) denotes a fully connected layer with a input neurons and b output neurons activated by function f, none denotes no activation function is used, and Drop(p) denotes an exit layer with probability p maintained during training.

The experiments were performed using only normal class data instances for model training according to the settings in the DSEBM, with training periods of 200 and 400 for KDDCUP and KDDCUP-Rev, respectively. The batch gradient descent was set to 1024. The training set and test set were split 1:1 and randomly selected.

*4.2. Performance Comparison*

After 20 runs of the DT-DAGMM and the baseline model, the mean precision, recall, and F1-measure scores of the evaluation metrics were compared visually in table format (Table 2).

**Table 2.** Experimental results.

| Method | KDDCUP | | | KDDCUP-Rev | | |
|---|---|---|---|---|---|---|
| | Precision | Recall | F1-Measure | Precision | Recall | F1-Measure |
| OC-SVM | 0.7457 | 0.8523 | 0.7954 | 0.7148 | 0.9940 | 0.8316 |
| DSEBM-e | 0.8619 | 0.6446 | 0.7399 | 0.7863 | 0.7884 | 0.7874 |
| DSEBM-r | 0.8521 | 0.6472 | 0.7328 | 0.7003 | 0.7013 | 0.7008 |
| DT-DAGMM | 0.9597 | 0.9646 | 0.9621 | 0.9568 | 0.9631 | 0.9599 |

Table 2 shows the average precision, recall, and F1-measure of the DTOC-SVM, DSEBM-e, DSEBM-r, and DT-DAGMM after 20 runs. The OC-SVM had a high recall of 99.4% at KDDCUP-Rev, but overall, the DT-DAGMM had better performance than the baseline method. The average precision improved by 13.98% and 21.31% for KDDCUP and

KDDCUP-Rev, respectively, and recall improved by 24.32% for KDDCUP and KDDCUP-Rev by 14% and 16% for the F1-measure, respectively.

The experimental results show that the anomaly detection algorithm proposed in this paper has better detection accuracy compared to traditional methods and deep learning methods.

## 5. Case Analysis

The coal mine has a mining depth of 400 m, has adopted mechanized comprehensive coal mining technology, and has completed the construction of some automatic monitoring systems, such as video monitoring, electromagnetic radiation, microseismic stress and temperature multi-field sensor monitoring, and personnel positioning. The coal mine has also improved the disaster monitoring system and data platform construction; realized the monitoring, collection, analysis, and alarm functions of disaster data and uploaded them in real time; and built a coal dynamic hazard early warning platform. Its system function architecture is shown in Figure 2.

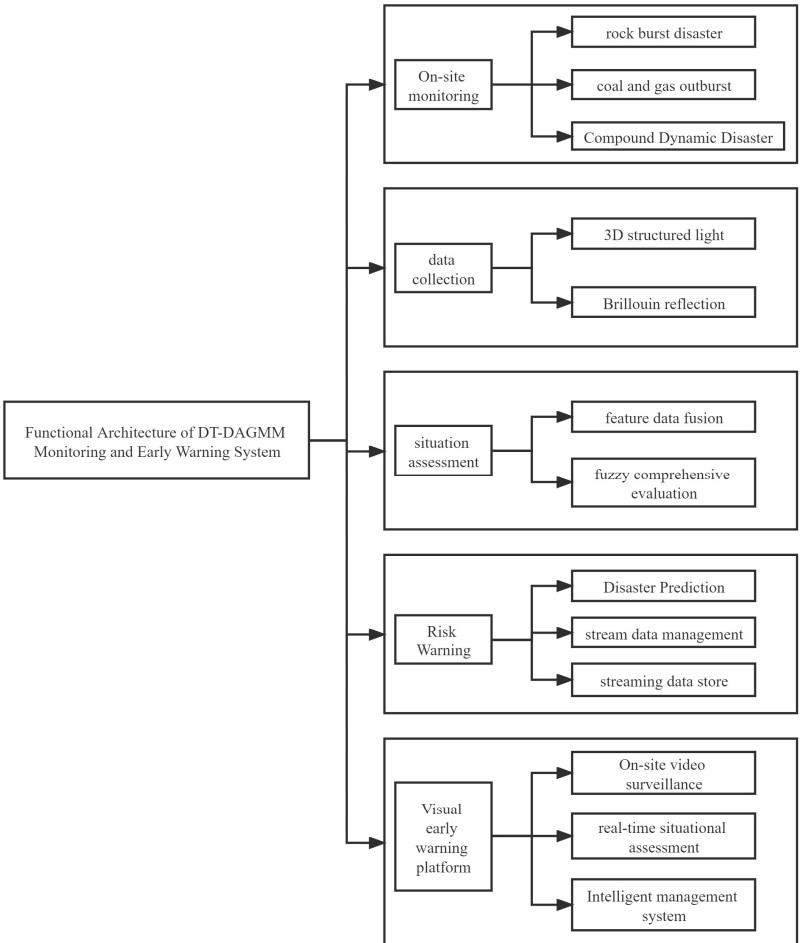

**Figure 2.** Prototype system functional architecture.

## 6. Conclusions

In order to improve the accuracy of coal-rock dynamic hazard anomaly detection, this paper proposed an anomaly detection method based on a dynamic threshold and a depth self-coding Gaussian hybrid model, which can quickly process high-dimensional data and solve the uncertainty of manually set thresholds. The experimental results show that the method has better performance. The research results of this paper provide a new detection

method based on the cross-disciplinary theory for coal-rock dynamic hazards, which is an important contribution to improving the reliability and accuracy of detection.

**Author Contributions:** Project administration, Z.L.; supervision, Z.L.; writing—original draft, Y.K. All authors have read and agreed to the published version of the manuscript.

**Funding:** This research received no external funding.

**Institutional Review Board Statement:** Not applicable.

**Informed Consent Statement:** Not applicable.

**Data Availability Statement:** The data used to support the results of this study are available from the authors upon request.

**Conflicts of Interest:** The authors declare no conflict of interest.

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
