# Peer review of "Improving the Accuracy of Coal-Rock Dynamic Hazard Anomaly Detection Using a Dynamic Threshold and a Depth Auto-Coding Gaussian Hybrid Model"

_sustainability, doi:10.3390/su15129655_

Round 1
Reviewer 1 Report
An anomaly detection method based on dynamic threshold and depth self-coding Gaussian hybrid model was proposed in this study. The comments to the authors for the revision and improvement are listed as follows.
(1) Coal-rock dynamic disaster is a kind of complex occurrence, which may be due to the stress concentration (ref to https://doi.org/10.1142/S1758825123500369), geotechnical structure damage (ref to https://doi.org/10.1016/j.ijmst.2022.11.006) and environmental factors (Unloading behaviours of shale under the effects of water through experimental and numerical approaches). The authors are suggested to add the information in the introduction part for better understanding.
(2) Some advanced technologies for early warning of dynamic disasters are suggested to be discussed in the second paragraph, such as the fractal analysis of time series (ref to An early-warning method for rock failure based on Hurst exponent in acoustic emission/microseismic activity monitoring). The advantages of these methods are also suggested to be added.
(3) As authors stated that the complexity of the geological environment and the high difficulty of data processing have made this world-wide problem unsolved so far, the uncertainty is the key reason. Uncertainty is an intrinsic feature in rock engineering, including the parameters and model determination (A practical and efficient reliability-based design optimization method for rock tunnel support). The uncertain analysis may be discussed.
(4) The first appearance of the abbreviation should be made in its full name.
(5) A case study of detection is suggested to be added in details, both using figures and description.
(6) What is the error analysis?
Author Response
Response to Reviewer 1 Comments
Point 1: (1) Coal-rock dynamic disaster is a kind of complex occurrence, which may be due to the stress concentration (ref to https://doi.org/10.1142/S1758825123500369), geotechnical structure damage (ref to https://doi.org/10.1016/j.ijmst.2022.11.006) and environmental factors (Unloading behaviours of shale under the effects of water through experimental and numerical approaches). The authors are suggested to add the information in the introduction part for better understanding.
Response 1: I have added the explanation requested by the reviewer in the Introduction section.
Point 2: Some advanced technologies for early warning of dynamic disasters are suggested to be discussed in the second paragraph, such as the fractal analysis of time series (ref to An early-warning method for rock failure based on Hurst exponent in acoustic emission/microseismic activity monitoring). The advantages of these methods are also suggested to be added.
Response 2: I added the advanced disaster warning technology to the second paragraph, and added the relevant advantages and disadvantages analysis.
Point 3: As authors stated that the complexity of the geological environment and the high difficulty of data processing have made this world-wide problem unsolved so far, the uncertainty is the key reason. Uncertainty is an intrinsic feature in rock engineering, including the parameters and model determination (A practical and efficient reliability-based design optimization method for rock tunnel support). The uncertain analysis may be discussed.
Response 3: I have added an analysis of uncertainty to the article to enrich my argument.
Point 4: The first appearance of the abbreviation should be made in its full name.
Response 4: I checked the full text and added the full name with abbreviations.
Point 5: A case study of detection is suggested to be added in details, both using figures and description.
Response 5: I added a case study chapter.
Point 6: What is the error analysis?
Response 6: There is no error analysis in the article, but the analysis of disaster prediction..

Reviewer 2 Report
The comments can be found in the attached file.

The English language of the text is enough for publication.
Author Response
Point 1: Please check the line 22 of the abstract section of the study. It is unclear.
Response 1: I have reworked the Abstract section to better illustrate the conclusions of the paper.
Point 2: Compared to previous approaches, what is the new achievement of the current study?
Response 2: This paper uses the DT-DAGMM algorithm, which has more accurate early warning in coal mine disaster early warning, making modern smart coal mines more advanced in disaster early warning.
Point 3: Compared to other strategies, what are the benefits of the DT-DAGMM model?
Response 3: At the end of the article, I compared several early warning algorithms and obtained the advanced nature of this algorithm.
Point 4: Please discuss on the applied strategy in this approach. Dose it improve the accuracy of the threshold?
Response 4: I have added related discussions in the article, and added the architecture and case analysis of the real system to the end of the article, and the conclusions can support its more advanced prediction accuracy.
Point 5: Compared to previous traditional methods, what is the advantage of the proposed laboratory data in the current approach?
Response 5: In this article, I compared the performance parameters of several algorithms. This method has indeed improved the prediction accuracy of the threshold, and it has been applied to the smart coal mine system and achieved good results.
Point 6: In this regard, the following studies https://doi.org/10.1016/j.ast.2017.06.008, https://doi.org/10.1016/j.ast.2018.03.010 should be also investigated.
Response 6: I have added two articles to the review section.
Point 7: Conclusion part of the study should be expanded remarking the achievements of the new study.
Response 7: The Discussion in the Conclusions section has been optimized as suggested by the comments.

Round 2
Reviewer 1 Report
None
None
Author Response
Dear Reviewer:
Thank you very much for your meticulous examination and valuable comments, which are very helpful to improve the quality of the article. I am very impressed by your rigorous work attitude and dedication, and I would like to express my deep respect.
I have already revised the English language according to your suggestions.
Thank you again for your hard work and help!
